



# Quarterdiurnal signature in sporadic E occurrence rates and comparison with neutral wind shear

Christoph Jacobi[1], Christina Arras[2], Christoph Geißler[1], and Friederike Lilienthal[1]

[1]Institute for Meteorology, Universität Leipzig, Stephanstr. 3, 04103 Leipzig, Germany
[2]Helmholtz Centre Potsdam German Research Centre for Geosciences - GFZ, Section 1.1: Space Geodetic Techniques, Telegrafenberg, 14473 Potsdam, Germany

**Correspondence:** Ch. Jacobi (jacobi@uni-leipzig.de)

**Abstract.** The GPS radio occultation (RO) technique is used to study sporadic E ($E_S$) layer plasma irregularities of the Earth's ionosphere on a global scale using GPS signal-to-noise ratio (SNR) profiles from the COSMIC/FORMOSAT-3 satellite. The maximum deviation from the mean SNR can be attributed to the height of the $E_S$ layer. $E_S$ are generally accepted to be produced by ion convergence due to vertical wind shear in the presence of a horizontal component of the Earth magnetic field, while the wind shear is provided mainly by solar tides. Here we present analyses of quarterdiurnal (QDT) signatures in $E_S$ occurrence rates. We find from a local comparison with mesosphere/lower thermosphere wind shear obtained with a meteor radar at Collm ($51.3°N$, $13.0°E$), that the phases of the QDT in $E_S$ agree well with those of negative wind shear for all seasons except for summer, when the QDT amplitudes are small. We also compare the global QDT $E_S$ signal with numerical model results. The global distribution of $E_S$ occurrence rates qualitatively agrees with the modeled zonal wind shears. The results indicate that zonal wind shear is indeed an important driving mechanism for the QDT seen in $E_S$.

## 1 Introduction

In the lower E region of the ionosphere, thin layers of high electron density are frequently found, the so called lower ionospheric sporadic E ($E_S$) layers. $E_S$ layers are thin clouds of accumulated plasma, which occur primarily at midlatitudes and maximize during summer (e.g., Arras et al., 2008). They are generally formed at altitudes between $90\,\text{km}$ and $120\,\text{km}$, i.e. in the lower thermosphere/lower ionospheric E region. According to wind shear theory (Whitehead, 1961) the process of $E_S$ formation is an interaction between the Earth's magnetic field, the ion concentration, and the vertical wind shear. Neglecting diffusion, the vertical velocity component of the neutral gas, and the electric force, the vertical ion drift $w_I$ may be written as (e.g., Haldoupis, 2012; Fytterer et al., 2014; Oikonomou et al., 2014):

$$w_I = \frac{r \cdot \cos I}{1+r^2}U + \frac{\cos I \sin I}{1+r^2}V, \tag{1}$$

where $U$ and $V$ are the zonal and meridional wind components of the neutral gas pointing towards east and north, resp., while $I$ is the inclination of the Earth's magnetic field. The parameter $r = \nu/\omega$ describes the ratio of the ion-neutral gas collision frequency $\nu$ and the gyro frequency $\omega = eB_0/m_I$, with $e$ as the elementary charge, $B_0$ as the total intensity of the Earth's

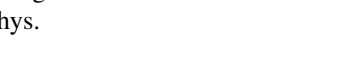


magnetic field, and $m_I$ as the ion mass. Note that in Eq. (1), in contrast to the usual notations in literature, Cartesian coordinates are used. Given that $r \gg 1$ below 115 km (Bishop and Earle, 2003), in the lower E region the zonal wind component is more efficient in causing vertical plasma motion than the meridional wind component. Consequently, the second term of (1) becomes small at that height, and therefore negative vertical zonal wind shear primarily leads to the formation of $E_S$. Note that

Eq. (1) holds only for magnetic midlatitudes (about 20° - 70°), where electric forces can be neglected. General correspondence between wind shear and $E_S$ had been found, e.g. from comparison using winds from the Horizontal Wind Model (Chu et al., 2014). Recently, Shinagawa et al. (2017) calculated the global distribution of the vertical ion convergence from GAIA Earth system model simulations and showed that their global distribution is roughly consistent with $E_S$ occurrence rates (OR), while Liu et al. (2018) found correspondence between $E_S$ OR taken from Global Positioning System (GPS) radio occultation (RO)

data and wind shear from the TIMED/TIDI satellite-borne zonal winds.

The dynamics of the lower thermosphere are strongly influenced by atmospheric waves, including the solar tides with periods of a solar day, and its harmonics. Their wind amplitudes usually maximize around or above 120 km. In these regions, tidal amplitudes are of the order of magnitude of the mean wind. Shorter period waves often have smaller amplitudes, so

that the main diurnal variability is owing to the diurnal tide (DT) and the semidiurnal tide (SDT), and also the terdiurnal tide (TDT). The quarterdiurnal tide (QDT), although it also forms an integral part of the middle and upper atmosphere dynamics, has attained much less attention, mainly due to its smaller amplitude. Assuming that solar tides are a major source of the vertical wind shear, frequently providing larger vertical gradients - both negative and positive depending on local time - than the background wind, tide-like structures are expected in $E_S$ occurrence rates. Actually, the SDT and DT are generally

accepted to be the major driver of $E_S$ (Mathews, 1998), leading to the reproduction of downward moving tidal signatures, e.g., in $E_S$ ionosonde registrations (Haldoupis et al., 2006; Haldoupis, 2012, e.g.,). By combining GPS $E_S$ registrations and radar wind measurements at midlatitudes, Arras et al. (2009) have shown that $E_S$ occurrence frequencies in the course of one day actually maximize when the zonal wind shear provided by the SDT is negative. Modeling by Resende et al. (2018b) showed the connection of $E_S$ and tides, although they focused in the equatorial region, where electric field effects become important.

More recently, Fytterer et al. (2013) found a clear correlation between midlatitude radar zonal wind shear and $E_S$ for the 8-hr component also. Fytterer et al. (2014) showed correspondence between the QDT in $E_S$ and wind shear on a global scale. Thus, Fytterer et al. (2013, 2014) confirmed that not only DT and SDT, but TDT wind shear as well contributes to $E_S$ formation.

There remains the question about the role of the QDT in the formation of $E_S$, and a possible connection with QDT neutral

wind shear at midlatitudes. Some publications report that no QDT have been found in some ionospheric records (e.g., Cyprus, 35°N, 33°E, Oikonomou et al., 2014). However, 6-hr tidal signatures have been observed in lower ionospheric $E_S$ parameters already (Tong et al., 1988; Morton et al., 1993). For the neutral atmosphere, observations of the QDT in barographic records (e.g. Warburton and Goodkind, 1977; Hupe et al., 2018) have been reported, and some reports on observations from mesosphere and lower thermosphere (MLT) radars are available (Smith et al., 2004; Liu et al., 2006; Jacobi et al., 2017, 2018; Guharay

et al., 2018). Few attempts to numerically model the QDT have been undertaken (Smith et al., 2004; Jacobi et al., 2018). On a



global scale, the 6-hr harmonics of ozone heating rates have been calculated from Aura/MLS satellite observations by Xu et al. (2012), who noted that the main 6-hr forcing during solstice is in the winter hemisphere. Xu et al. (2014) analysed nonmigrating tides from TIMED/SABER observations. In a further study, Liu et al. (2015), again using TIMED/SABER data, analysed the migrating QDT between 50°S and 50°N in the middle atmosphere. Azeem et al. (2016) analyzed temperature data from the

5    NIRS instrument on the International Space Station and from SABER during June and July 2010, and found that the QDT was a significant dynamical feature in the thermosphere. The seasonal/latitudinal structure of the QDT is complex; generally, the seasonal cycle exhibits a maximum in winter and also during equinoxes, and, considering the latitudinal distribution, several maxima at low, middle and higher latitudes (Smith et al., 2004; Liu et al., 2015; Azeem et al., 2016). The latitudinal structure is dominated by the (4,6) Hough mode, but other modes are also present (Liu et al., 2015).

Returning to $E_S$, their 6-hr component has not yet been analysed in detail using GPS RO observations, which motivates us to search for the QDT signature in $E_S$ OR derived from RO, and compare them with wind shear observations and model predictions. Therefore, in this paper we analyse the quarterdiurnal oscillation seen in $E_S$, obtained from GPS RO measurements by the FORMOsa SATellite mission-3/Constellation Observing System for Meteorology, Ionosphere and Climate (FORMOSAT-

15   3/COSMIC). We compare $E_S$ phases with phases of negative wind shear obtained from local radar observations at Collm (51.3°N, 13.0°E), and compare the global distribution of 6-hour amplitudes with the wind shear amplitudes from numerical modeling. The remainder of the paper is organised as follows. In section 2 the $E_S$ detection and the radar wind observations are briefly described, and the numerical global circulation model is introduced. Results of QDT analysis and comparison with wind shear observations and modeling are presented in section 3. Section 4 concludes the paper.

## 2   Dataset and model description

### 2.1   Sporadic E occurrence rates

The FORMOSAT-3/COSMIC constellation consists of six low-Earth orbiting (LEO) microsatellites which orbit the Earth at an initial altitude of  800 km. The satellites perform RO measurement in both the neutral atmosphere and the ionosphere (Anthes et al., 2008). During an occultation, signals of rising or setting GPS satellites are received by a LEO satellite. While the sig-

25   nals pass Earth's atmosphere they are modified by atmospheric conditions, in particular ionospheric electron density, causing refraction and degradation of the GPS waves, which can be utilized to obtain information about the ionosphere and neutral atmosphere. More detailed information on the principles of the RO technique is given by Hajj et al. (2002) and Kursinski et al. (1997).

30   The method to derive $E_S$ information from RO signals has been described in Arras and Wickert (2018). In brief, for our investigations, the Signal-to-Noise ratio (SNR) profiles of the GPS L1 phase measurements are used. The SNR is very sensitive to vertical variations of the electron density, and these occur within an $E_S$ layer. These vertically localized electron density variations lead to phase fluctuations of the GPS signal, which can be observed as changes of the received signal strength (Hajj



et al., 2002). In order to avoid influences from the different basic signal power values on the further data analysis, every SNR profile is normalized first. In the case of absence of ionospheric disturbances the SNR value is almost constant at altitudes above $35\,\mathrm{km}$. The SNR standard deviation profile is considered to be disturbed when it exceeds an empirically found threshold of 0.2. If large standard deviation values are concentrated within a layer of less than $10\,\mathrm{km}$ vertical extent, we assume that the

respective SNR profile includes the signature of an $E_S$ layer. The height where the SNR value deviates most from the mean of the SNR profile is considered as the altitude of the $E_S$ layer. This has been validated by comparisons with ionosonde $E_S$ observations (Arras and Wickert, 2018; Resende et al., 2018a).

Figure 1 shows 2007 - 2016 mean zonal mean $E_S$ OR. OR have been calculated as the number of $E_S$ within a $5°$ latitude

and $10\,\mathrm{km}$ height window, divided by the number of RO in the respective latitude window. Figure 1 shows seasonal means for December–February (DJF), March–May (MAM), June–August (JJA) and September–November (SON). The distributions are similar to those shown by Fytterer et al. (2014) obtained from a more limited dataset. Maximum OR are found at altitudes slightly above $100\,\mathrm{km}$. OR maximize in summer, which is thought to be owing to increased meteor influx during that season (Haldoupis et al., 2007). The summer maximum is more pronounced in the Northern Hemisphere, which is due to the South

Atlantic Anomaly and the weaker magnetic field there (e.g., Arras et al., 2008; Chu et al., 2014; Arras and Wickert, 2018), so that Southern Hemisphere summer zonal mean OR are smaller than Northern Hemisphere ones. Near the equator, $E_S$ OR are small, owing to the horizontal magnetic field at the magnetic equator, which does not allow electrons to follow the vertically moving ions (e.g., Arras et al., 2008, 2010; Arras and Wickert, 2018).

## 2.2 Collm mesosphere/lower thermosphere wind shear

At Collm ($51.3°$N, $13.0°$E), a SKiYMET meteor radar is operated on $36.2$ MHz since summer 2004. The radar operates in an all-sky configuration, and the main parameter observed are the MLT radial winds determined from the Doppler shift of individual meteor trails. Details of the radar system and the radial wind determination principle can be found in Jacobi (2012), Stober et al. (2012), and Lilienthal and Jacobi (2015). During 2015 the radar had been upgraded by increasing the peak power, and replacing the Yagi antennas by crossed dipoles. The transmit frequency is still the same (Stober et al., 2017). The individual

meteor trail reflection heights vary between about 75 and 110 km, with a maximum meteor count rate slightly below 90 km (e.g., Stober et al., 2008). The data are binned here in 6 different not overlapping height gates centered at 82, 85, 88, 91, 94, and 98 km. The hourly mean reflection height may slightly deviate from the nominal heights due to the uneven height distribution of meteors within each gate (Jacobi, 2012). Individual radial winds calculated from the meteors are collected to form hourly mean values using a least squares fit of the horizontal wind components to the raw data under the assumption that vertical

winds are small (Hocking et al., 2001). Hourly values of the vertical shear of zonal wind (subsequently 'zonal wind shear' in brief) have been calculated from adjacent height gates as in Arras et al. (2009) and Fytterer et al. (2013), and the reference height for shear values has been attributed to the center between the nominal heights of the wind values.



An example for the diurnal wind and shear variation in the MLT over Collm is given in Figure 2. On the left panel, height-time cross-sections of 2007 - 2016 mean DJF mean diurnal zonal winds and wind shears over Collm are shown. Maximum wind values exceed 70 (50) ms$^{-1}$ in eastward (westward) direction, while the zonal wind shear maximizes at more than $\pm 8$ ms$^{-1}$, similar to the values shown by Arras et al. (2009). Clearly, the main contribution to zonal wind and wind shear variability in

winter is due to the SDT. On the right panel of Figure 2, the diurnal zonal wind shear values $S(t)$ at 92.5 km are shown, together with a modeled least-squares fit including mean $S_0$, 8 hr, 12 hr, and 24 hr components:

$$S_{Mod}(t) = S_0 + \sum_{i=1}^{3} a_i \sin \frac{2\pi}{P_i} t + b_i \cos \frac{2\pi}{P_i} t, \qquad (2)$$

with $t$ as the time and $P_i$ as the above mentioned periods, and the coefficients $a_i$ and $b_i$ being determined through minimizing $\sum (S(t) - S_{Mod}(t))^2$. The amplitudes $A_i$ and phases $T_i$ of the wind shear are calculated as

$$A_i = \sqrt{a_i^2 + b_i^2}, \qquad T_i = \frac{P_i}{2\pi} \arctan \frac{a_i}{b_i} + \frac{P_i}{2}. \qquad (3)$$

Note that the phases are defined here as the time of maximum negative wind shear, so that a $P_i/2$ - term is added to the right part of (3). The residuals $S(t) - S_{Mod}(t)$, multiplied by a factor of 5 for better visibility, are added as blue line on the right panel of Figure 2. Obviously, there is a quarterdiurnal signature during this season and at this altitude level. Amplitudes and phases of this variation are calculated via a least-squares fit similar to Eqs. (2) and (3) but in addition including the QDT period

$P_4 = 6$ hr in the analysis.

Figure 3 shows the 2007 - 2016 mean seasonal mean Collm residual wind shear after removing mean shear, 8 hr, 12 hr, and 24 hr components for (a) DJF, (b) MAM, (c) JJA, and (d) SON. 6-hr phases, defined as the time of maximum negative wind shear according to Eq. (3), are added. The QDT at Collm is relatively strong in winter, but very weak in summer (Jacobi et al.,

2017), and consequently the 6-hr signal is not visible for JJA. For the other seasons the QDT shear tends to increase with altitude, although in MAM the QDT signal is not the major one at the upper height gate. In SON, the seasonal means consist of a superposition of summer, transition, and winter oscillations (see Jacobi et al., 2017, their Figures 3 and 6). As a consequence, at the upper height gates a clear QDT signature is visible, but these are not connected with the QDT at the lower heights, also leading to a break in the vertical phase change.

### 2.3 MUAM circulation model predictions

We use the nonlinear Middle and Upper Atmosphere Model (MUAM) to investigate the QDT with wavenumber 4. MUAM is a 3-dimensional mechanistic model based on the COMMA-LIM (Fröhlich et al., 2003b; Jacobi et al., 2006) model. The more recent version of the model, MUAM, is documented by Pogoreltsev (2007), Pogoreltsev et al. (2007), and Lilienthal et al. (2017,

2018). MUAM extends from the surface (1000 hPa) to the lower thermosphere while the lower 30 km zonal mean temperatures are nudged with monthly mean 2000-2010 mean ERA-Interim reanalyses of zonal mean temperature. This ensures that the



zonal mean dynamics of the lower atmosphere is close to the reanalyses, while waves are allowed to form anyway. The model has a horizontal resolution of $5 \times 5.625°$ and a vertical resolution of $2.842\,\text{km}$ in logarithmic pressure height with a constant scale height of $H = 7\,\text{km}$. Parameterizations of gravity waves, solar, and infrared radiation as well as several ionospheric effects are included, the latter, however, are only represented based on simple empirical distributions.

The parameterization of solar heating in the middle atmosphere is calculated following Strobel (1978). It considers heating due to the most important gases such as $H_2O$, $CO_2$, ozone, $O_2$, and $N_2$. Note that these gases are taken as zonal means, different from other versions of MUAM (e.g., Suvorova and Pogoreltsev, 2011; Ermakova et al., 2017), so that mainly migrating tides are forced through solar heating. Monthly mean zonal mean ozone fields up to $50\,\text{km}$ altitude are taken from the Stratosphere-troposphere Processes And their Role in Climate project (SPARC; Randel and Wu, 2007), and an exponential decrease of ozone is applied above 50 km. Monthly volume mixing ratios for $CO_2$ have been chosen according to measurements from Mauna Loa Observatory for the year 2005 (e.g., $378\,\text{ppm}$ for January; NOAA ESRL Global Monitoring Division), and the mixing ratio is assumed constant across latitudes and longitudes. $CO_2$ are taken as constant with height until 87.5 km, and then decrease exponentially. Chemical heating due to recombination of $O_2$ and $O$ (Riese et al., 1994), and heating due to extreme ultraviolet radiation are added. This is described in more detail by Fröhlich et al. (2003a).

Gravity waves in the middle atmosphere are calculated by a linear Lindzen-type (Lindzen, 1981) parameterization based on Jakobs et al. (1986) and updated as described by Fröhlich et al. (2003a, b) and Jacobi et al. (2006). Due to the fact that this parameterization does not account for gravity waves in the thermosphere, it is coupled with a modified parameterization after Yiğit et al. (2008), connected via the eddy diffusion coefficient. To avoid large interactions between both parameterizations, we limit the phase speeds of the Lindzen-type scheme to $5 - 30\ \text{ms}^{-1}$ , while the Yiğit-scheme covers larger phase speeds between $35 - 105\ \text{ms}^{-1}$ . This way, the Yiğit parameterization mainly attributes the thermosphere while the Lindzen-type parameterization affects the stratosphere and mesosphere. Overlaps between both parameterizations are small and the contributions of both routines to the tendency terms can be simply summed up (Lilienthal et al., 2018).

In the configuration used here, the model incorporates a spin-up of 120 model days. Within that time, heating rates are zonally averaged and are therefore building up a background climatology without any tidal forcing. Note that we do not explicitly excite any kind of waves at the lower boundary during the whole simulation. In the following 90 model days, heating rates are allowed to be zonally variable and tides start to propagate. In this model version, the sun's zenith angle does not account for day-to-day variations and refers to the middle of the respective month. The last 30 model days are analyzed and presented here. Since there is no change of any forcing, the day-to-day variability is negligible. Figure 4 shows the annual cycle of the simulated monthly 6-hr zonal wind amplitudes at $101\,\text{km}$ altitude. The amplitudes are larger in winter than in summer, with winter maximums at higher and middle latitudes, which have also been observed (Jacobi et al., 2017), and is also predicted by Smith et al. (2004). There is also a tendency for the spring maximum at midlatitudes as reported by Jacobi et al. (2017). We also note maximums at 30 - 40°, as has been reported by Liu et al. (2015). Note that, since the solar forcing is parameterized





using zonal mean climatologies of minor species, the MUAM amplitudes only show migrating components, while the radar observations only deliver the total amplitude. However, from amplitude and phase comparisons between two stations, Jacobi et al. (2017) concluded that the major contribution to the QDT observed by radar is due to the migrating tide, and comparison between model results and radar is justified.

## 3 Results

### 3.1 Local comparison of 6-hr radar wind shear and $E_S$ occurrence rates

Here, we compare the diurnal cycle of $E_S$ OR at the latitude of Collm with the wind shear observed by the MR with respect to the QDT signature, similar to the approach of Arras et al. (2009) and Fytterer et al. (2013) for the 12 and 8 hr component, respectively. Similar as in Figure 1, OR have been calculated as the number of $E_S$ divided by the number of RO, but here a latitude window of $10°$ centered at $51°$N was chosen, and the data were sampled in hourly bins according to local time. Due to this sampling irrespective of longitude, only migrating components contribute to the diurnal cycle in our analyses. As in Figure 1, the data refer to $10$ km height gates. The diurnal cycles for four seasons are shown in Figure 5. We note the downward propagation of $E_S$ OR signatures, which are dominated by the SDT and, to a lesser degree, by a diurnal variation (see, e.g., Arras et al., 2009). The strongest amplitudes are seen in summer, while minimum amplitudes are found in winter, which is owing to the overall seasonal cycle of OR at higher midlatitudes, see Figure 1.

The residuals of $E_S$ OR after removing daily mean, $8$ hr, $12$ hr, and $24$ hr components, i.e. calculated in the same manner as the wind shear residuals in Figure 3, are shown in Figure 6. The QDT phases are calculated according to the right part of Eq. (3), but without the $P_4/2$-term, since we are interested in maximums of $E_S$. Zonal wind shear phases taken from Figure 3 are also shown in the lower parts of the respective panels. Again, solid symbols indicate that the amplitudes are significant at the $5\%$ level according to a $\hat{t}$-test. The QDT in $E_S$ is strongest in summer, as are the overall OR (see Figure 5). Vertical phase gradients in summer are large, but smaller in winter and autumn, which is also the case with the QDT in neutral winds (Smith et al., 2004; Jacobi et al., 2017). In autumn, significant QDT amplitudes are only found in a small height range at $95 - 100$ km. Generally, the QDT in $E_S$ disappears below $90 - 95$ km. Except for summer, there is a clear correspondence between negative wind shear phases and $E_S$ phases in the upper two radar height gates, i.e. above $90$ km. This indicates, that the QDT in $E_S$ actually forms at the nodes of the negative QDT wind shear component, and the $E_S$ formation process, responsible for the strong $12$ hr-component in $E_S$, also acts for the much weaker QDT.



### 3.2 Global distribution of 6-hr wind shear and $E_S$ occurrence rates

The $E_S$ QDT amplitudes are not necessarily directly related to the respective wind shear amplitudes, because $E_S$ intensities and OR depend on wind shear, but also on the ionization of metallic ions, and the latter are thought to exhibit a seasonal cycle owing to the variability of meteor influx (Haldoupis et al., 2007). Therefore, e.g. at midlatitudes largest $E_S$ OR amplitudes are

found in summer (see Figure 6), while the largest QDT in the neutral wind is seen during winter (Figures 3 and 4). Further, wind shear theory includes the influence of the horizontal magnetic field leading to a latitudinal dependence of $E_S$ OR, as well as to anomalies like the South Atlantic Anomaly. To take this into account when showing the global distribution of $E_S$ QDT amplitudes, following Fytterer et al. (2014) we calculated relative amplitudes $A_{4,r} = A_4/\mathrm{OR}$. We show their global distribution at an altitude of $101\,\mathrm{km}$ in Figure 7a. The zonal wind shear QDT at that height, calculated from MUAM monthly simulations

is shown in Figure 7b. Both parameters show a corresponding maximum in the winter higher midlatitudes hemisphere. In the Southern Hemisphere, the QDT in both $E_S$ and wind shear is small during boreal winter. In the Northern Hemisphere summer, a further QDT maximum in $E_S$ is found, which is, however, only very weakly represented in the wind shear. During the equinoxes, again corresponding maximums are found at lower and middle latitudes, especially on the Northern Hemisphere. Overall, there is a striking similarity between $E_S$ and wind shear QDT, in particular taking into account that the wind shear

here is not taken from observations, but from numerical modeling with tides being forced self-consistently and not based on observed distributions.

Figure 8 shows on the left column seasonal mean relative $E_S$ amplitudes as latitude-height plots, while the corresponding modeled QDT shear amplitudes are presented on the right column. We show values only between 90 and $100\,\mathrm{km}$, because

below and above the OR become small (Figures 1 and 5), and therefore the $A_{4,r}$ distribution tends to become more irregular. There is an overall close correspondence between $E_S$ and wind shear during the solstices (panels a,b and e,f). Wind shear amplitudes then are large on the middle to high latitude winter hemisphere, with large values down to the upper mesosphere, while corresponding $E_S$ amplitudes are large there, too. A secondary maximum in the lower thermosphere winter near $30°$ is also seen in both $E_S$ and wind shear. In JJA, there is also a corresponding maximum on the summer hemisphere. This is

also visible in the DJF wind shear, however, only very weakly expressed in $E_S$. During boreal spring (panels b, c), there are corresponding maximums in $E_S$ and shear QDT at higher midlatitudes, maximizing at $60°$S and at $45$ - $60°$N, and also an indication for a joint maximum close to the equator. The latter, however, is probably coincidental, since the wind shear theory according to Eq. (1) does not hold for magnetic latitudes below about $20°$. There is a modeled QDT wind shear maximum near $30°$S during MAM, which is not visible in $E_S$, similar to the situation during austral spring (panels g,f). Generally, the

correspondence between wind shear and $E_S$ QDT is weakest during SON. This may partly be due to the seasonal cycle and the fact that seasonal averages are taken from monthly data that may be very different (e.g., a clear minimum/maximum in September/November at higher northern midlatitudes, see Figure 7). Also, the overall correspondence between $E_S$ and wind shear QDT mainly refers to the position of maximums and minimums, while the absolute values may differ. Still, however, there is a correspondence visible between the global distribution of the QDT in $E_S$ and the one seen in wind shear, indicating



the presence of the wind shear mechanism also for the QDT.

## 4    Conclusions

We have analyzed the migrating QDT amplitudes and phases from $E_S$ OR analyzed from GPS RO. Comparing the phases
with those from wind shear observed by radar over a midlatitude site shows a clear correspondence, indicating that the wind
shear mechanism is indeed an important source for the $E_S$ QDT. Note, however, that we have compared the local QDT phases
with the migrating signal from $E_S$, and although the migrating QDT is the dominating one at the latitude of the observations,
a future analysis will have to consider this in detail, and also analyze the nonmigrating components from the $E_S$ QDT. Wind
shear and $E_S$ QDT phase gradients are smaller in winter than in summer, indicating shorter QDT wavelengths in summer than
in winter, which is in agreement with observations (Smith et al., 2004; Jacobi et al., 2017). However, as long as the amplitudes
do not increase exponentially, phase gradients also include the effect of amplitude change, and wavelengths cannot be derived
from the shear phases directly, so that the global $E_S$ OR will not provide the neutral atmosphere wavelengths, but only deliver
a qualitative measure.

Amplitudes of tidal signatures in $E_S$ are not only determined by the wind shear, but also depend on ion concentration and
Earth's magnetic field parameters, so that a correlation of $E_S$ and wind shear is possible only for a defined region, and for each
season separately. Comparing the amplitude distributions on a global scale was possible by dividing the $E_S$ amplitudes by the
background OR, which will take out most of the seasonal and regional dependencies of $E_S$ OR. Indeed, the global structure of
the QDT in $E_S$ OR and wind shear show strong similarities, which, besides the indication that the wind shear mechanism is
actually an important driver for $E_S$ formation, gives some confidence in the horizontal structure of the modeled tides also.

The modeled amplitudes are too small, compared with observations. Although it is an ongoing question that numerical mod-
els tend to underestimate tides, at least for some regions or seasons (e.g., Smith, 2012; Pokhotelov et al., 2018), the reasons
for this underestimation in MUAM is subject to current investigations. Another issue is the relatively coarse meridional model
resolution, which may smooth some details of the meridional structure of the QDT. Future experiments will be performed with
increased resolution.

*Code availability.*  The MUAM model code can be obtained from the corresponding author on request.

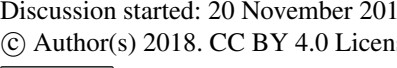


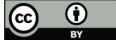

*Data availability.* Radio occultation data are freely available from UCAR on http://cdaac-www.cosmic.ucar.edu/cdaac/products.html. Collm radar wind shears are available from the corresponding author on request.

*Author contributions.* C. Jacobi performed Collm radar wind measurements and analyses, as well as the tidal analyses based on GPS $E_S$, which had been analyzed by C. Arras. C. Geißler designed and performed the MUAM model runs together with F. Lilienthal. C. Jacobi

5 drafted the first version of the text.

*Competing interests.* C. Jacobi is one of the Editors-in-Chief of Annales Geophysicae.

*Acknowledgements.* The provision of FORMOSAT-3/COSMIC data by University Corporation for Atmospheric Research is gratefully acknowledged. Ch. Jacobi, F. Lilienthal, and C. Geißler acknowledge support through the Deutsche Forschungsgemeinschaft (DFG) under grants JA 836/30-1 and JA 836/34-1. C. Arras acknowledges support by the DFG Priority Program DynamicEarth, SPP 1788.





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





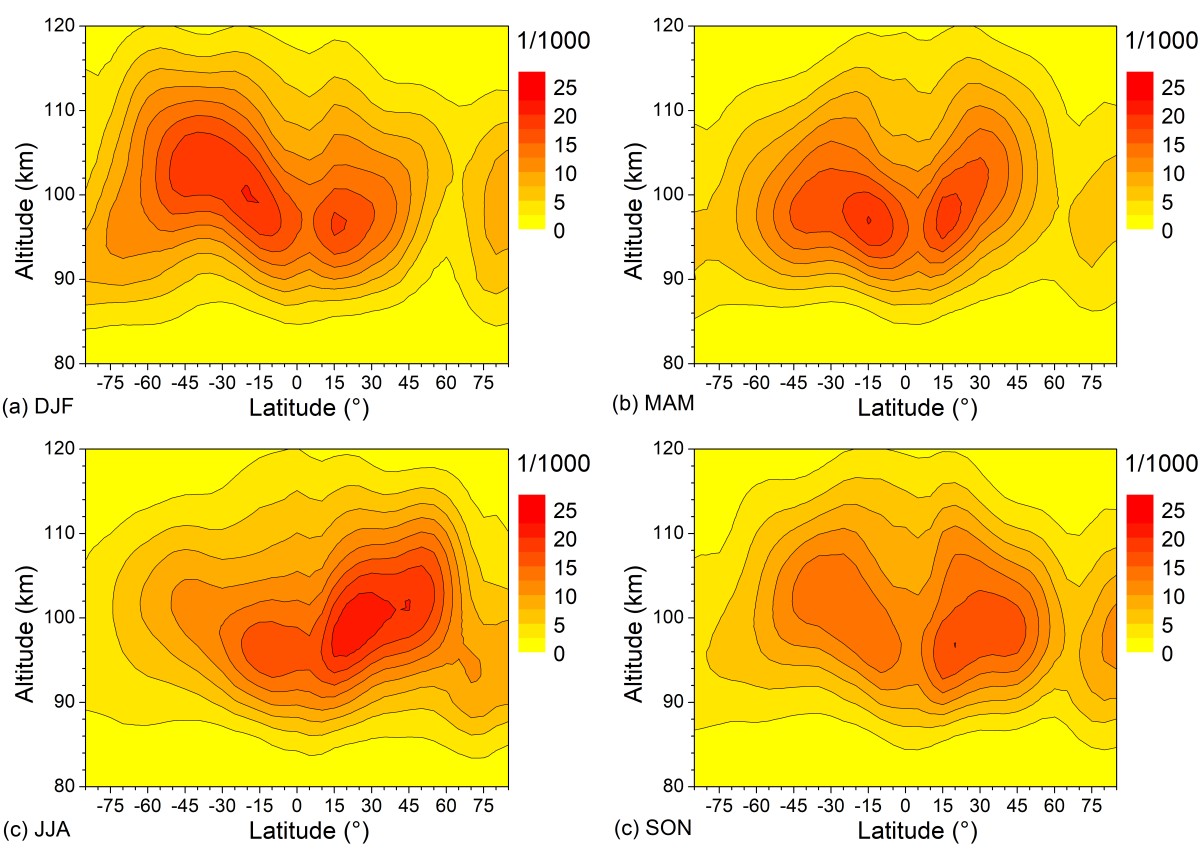

**Figure 1.** Zonal and seasonal mean $E_S$ occurrence rates for (a) DJF (b) MAM (c) JJA (d) SON. Data are averages over 2007 - 2016.



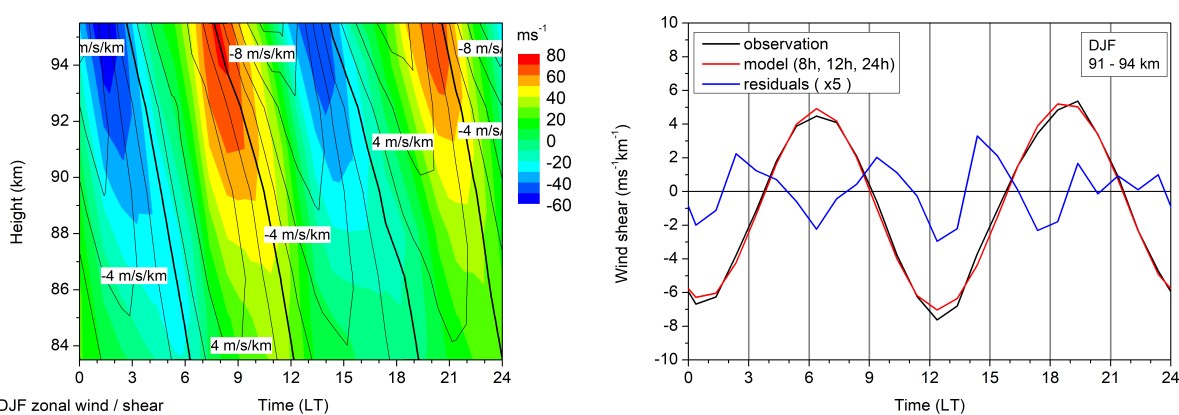

**Figure 2.** 2007 - 2016 mean winter (December - February) mean diurnal cycle of zonal wind and wind shear over Collm. Left panel: Height-time cross-sections of zonal wind (color coding) and wind shear (contour lines). Right panel: Wind shear at about 91.5 km (black line), together with a fit including mean, 8 hr, 12 hr, and 24 hr components (red line). The residuals, multiplied by a factor of 5, are added as blue line.



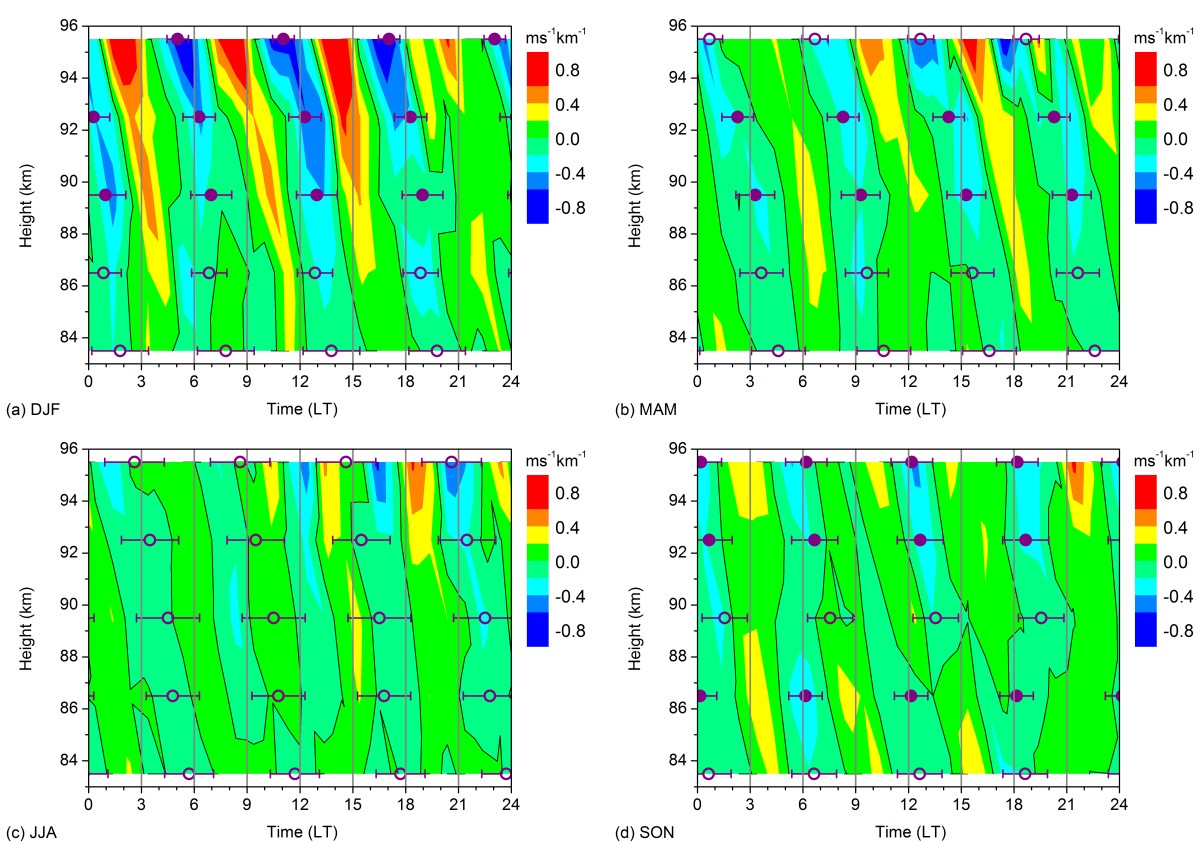

**Figure 3.** 2007 - 2016 mean seasonal mean Collm residual wind shear after removing diurnal mean shear, 8 hr, 12 hr, and 24 hr components for (a) DJF, (b) MAM, (c) JJA, (d) SON. 6-hr phases, defined as the time of maximum negative wind shear, are added together with their standard deviation calculated from phases of single years. Solid symbols denote oscillations significant at the 5% level according to a t̂-test.



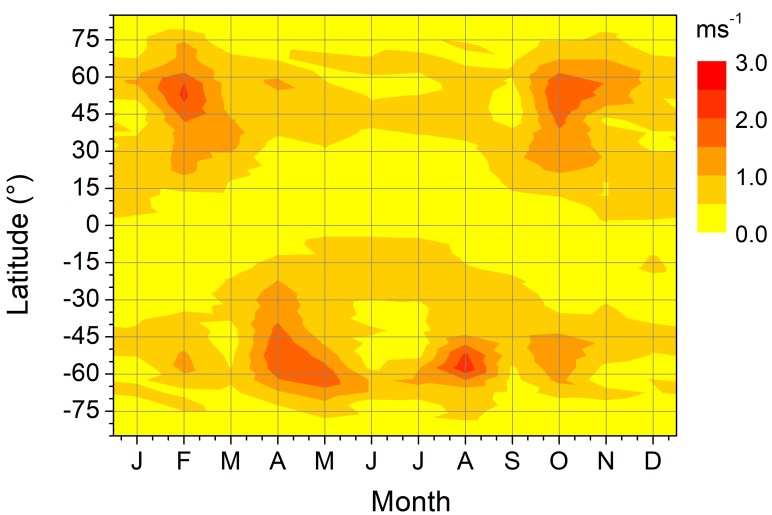

**Figure 4.** QDT zonal wind amplitudes at 101 km altitude, as modeled by MUAM.





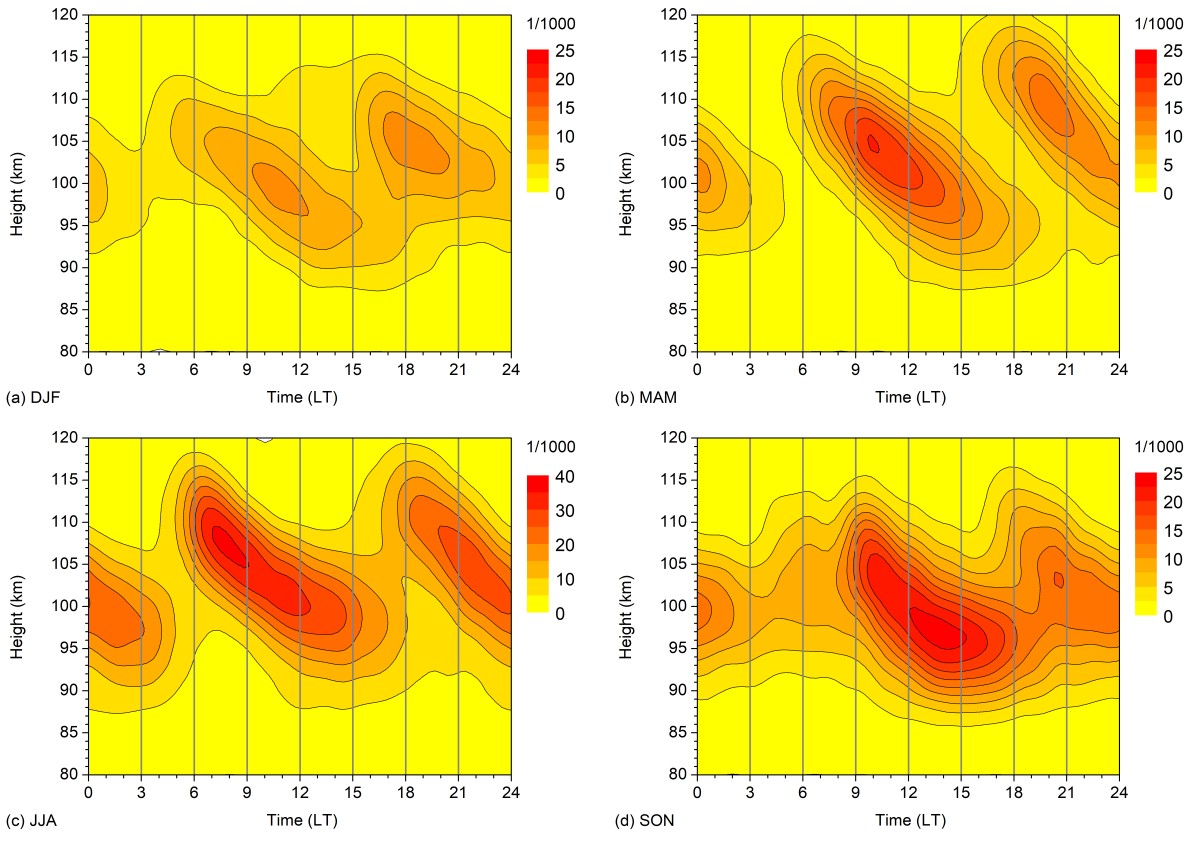

**Figure 5.** 2007 - 2016 mean seasonal mean diurnal cycle of $E_S$ occurrence rates for (a) DJF, (b) MAM, (c) JJA, (d) SON in a latitude band 46 - 56°N and in height gates of 10 km. Note the different scaling for JJA.





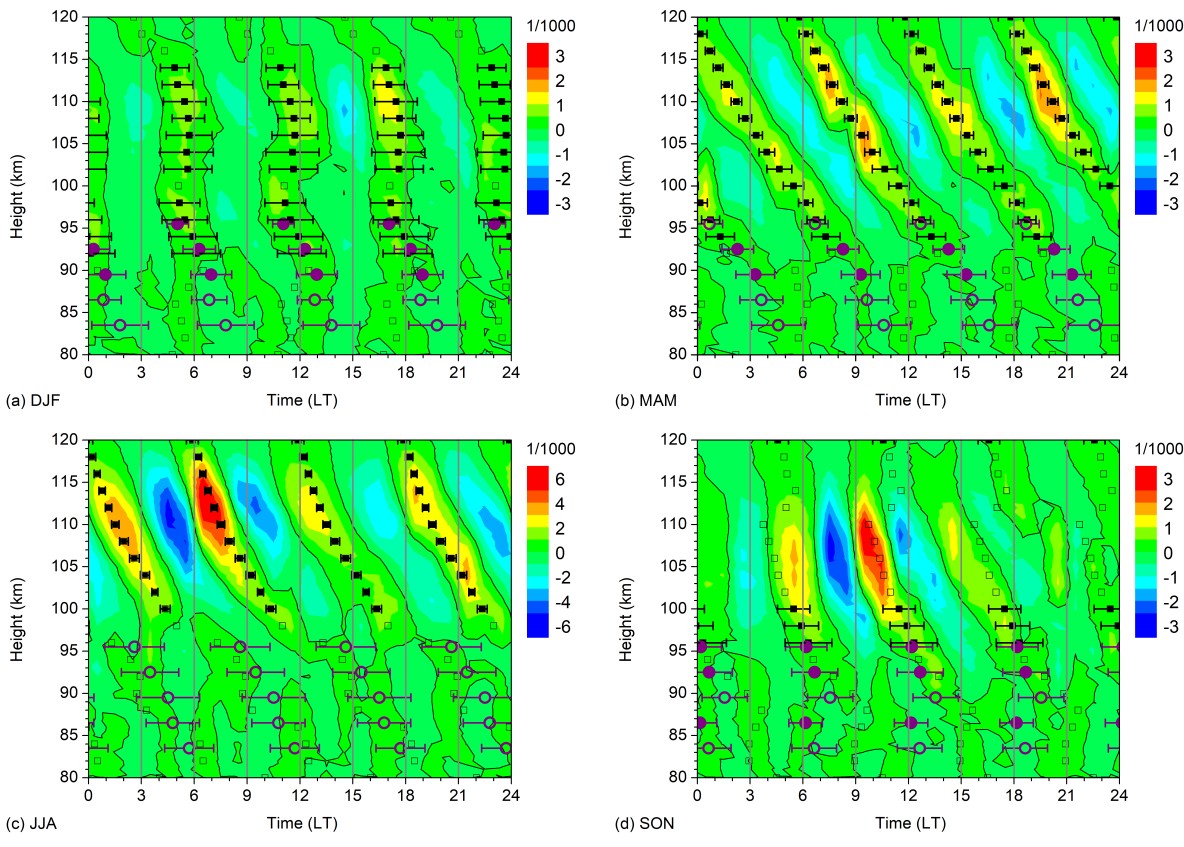

**Figure 6.** 2007 - 2016 mean seasonal mean diurnal cycle of $E_S$ residual occurrence rates after removing diurnal mean, 8 hr, 12 hr, and 24 hr components for (a) DJF (b) MAM (c) JJA (d) SON in a latitude band $46 - 56°$N and in height gates of $10$ km. Note the different scaling for JJA. 6-hr phases of OR and Collm wind shear are added as squares and circles, respectively. Solid symbols denote oscillations significant at the 5% level according to a $\hat{t}$-test. The error bars show standard deviations calculated from phases for single years.

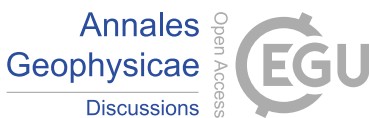



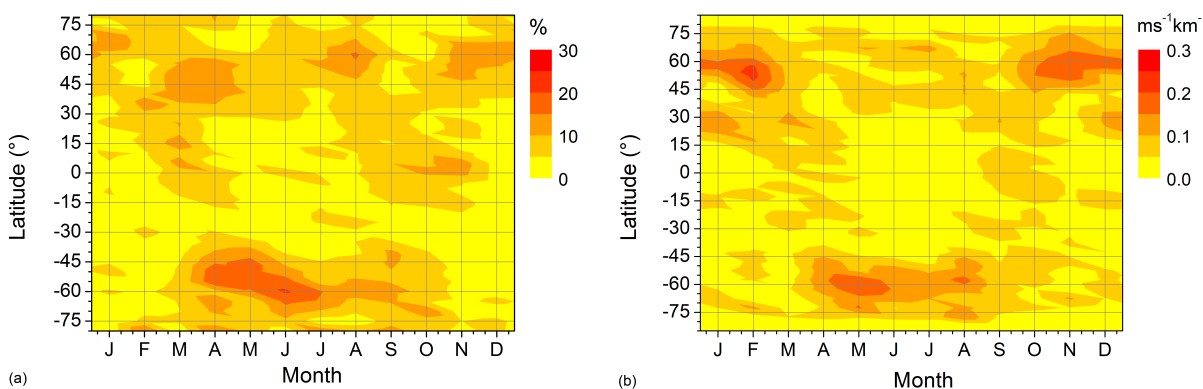

**Figure 7.** Left panel (a): 2007 - 2016 mean monthly mean QDT $E_S$ relative amplitudes at 101 km. Right panel (b): MUAM zonal wind shear at 101 km.



**Figure 8.** Left column (a): 2007 - 2016 mean monthly mean QDT $E_S$ relative amplitudes for (a) DJF), (c) MAM, (e) JJA, (g) SON. Right columns: corresponding MUAM QDT component of zonal wind shear.