# Peer review of "Quarterdiurnal signature in sporadic E occurrence rates and comparison with neutral wind shear"

_Annales Geophysicae, 2018_

## Referee Comment (RC2)

Review report of the manuscript entitled "*Quarterdiurnal signature in sporadic E occurrence rates and comparison with neutral wind shear*" by Jacobi et al.

The present work portrays the QDT signature in the $E_S$ plasma irregularities in the ionosphere and their relation with the MLT region neutral wind shear. They found good agreement between the positive phase of the QDT in the $E_S$ layers and the negative wind shear in the MLT most of the time of the year. Their results are interesting and well-presented in the manuscript. However, a few important points should be addressed before it can be considered for publication.

**Specific points**

- P-4, Line. 12-13: Authors mentioned that "slightly above 100 km". Actually, it is mostly below 100 km.

- P-7, Line. 11-12: "Due to this sampling irrespective of longitude, only migrating components contribute to the diurnal cycle in our analyses". It is not clear why only migrating components contribute.

- Figure 1a: In northern winter high OR near 15° latitude around 98 km is evident. What is the reason for that? What is the reason for equinoctial asymmetry in OR during SON while it is very symmetric during MAM?

- Figure 2: Only winter is shown. Results of all 4 seasons could be shown in order to be complete and consistent with Figure 1.

- Figure 4: Only a single representative altitude (110 km) is shown. More altitude bins (3 or more) could be included to strengthen the claim of the authors.

- Figure 6: There is a clear inconsistency in the QDT phase between the occurrence rate and wind shear in summer when QDT is found to be weak as mentioned by the authors. It implies that the estimated phases in the wind shear in summer are incorrect. Authors should substantiate the statement in detail.

- Figure 7: Only 110 km altitude is selected. At least 3 more altitude bins could be shown to corroborate the inference.

**Technical points**

- P-1, Line. 5: "quarterdiurnal (QDT)" to "quarterdiurnal tide (QDT)"
- P-2, Line. 17-19: "Assuming…." Correct the sentence.
- P-2, Line. 24: Correct "connection of" to "connection between"
- P-2, Line. 27: add "and QDT" after "TDT"
- P-3. Line. 6-8: "The seasonal/latitudinal…….." Split the sentence, it is too long.
- P-9, Line. 10-13: The sentence is too long. It is better to split.
- P-9, Line-16: Correct "so that a correlation of" to "so that a correlation between"
- P-9, Line. 21: Correct "compared with observations" to "in comparison with observations"
- P-9, Line. 26: Correct "increased resolution" to "higher resolution"
- Figure 1: Colorbar: Correct "1/1000" to "×1/1000"

- Articles are missing at several places in the text.
- In "introduction" section to describe past studies, both present perfect and past tenses are used. Authors are suggested to adhere to past tense only.

---

## Referee Comment (RC1) · Anonymous Referee #1 · 7 Dec 2018

The paper brings interesting results and the quality is high. I have almost no comments and I recommend to publish it with one exception:

Could you please define meaning of "positive" and "negative" in the context of zonal and meridional wind shear? It is missing in the text so that the description of winds is not clear:

...therefore NEGATIVE vertical zonal wind shear primarily leads to... (page 2, line 4), and ... when the zonal wind shear provided by the SDT is NEGATIVE. (page 2, line 23)

---

## Author Comment (AC2) · 24 Jan 2019

We are thankful for the referee's comments which will help us to improve the paper. We will address the raised issues in the revised version. In response to the specific points, we will briefly address each one here and briefly reply:

1. "P-4, Line. 12-13: Authors mentioned that "slightly above 100 km". Actually, it is mostly below 100 km." Response: We agree that our statement is only true for higher latitudes. We will correct that in a revised version.

2. P-7, Line. 11-12: "Due to this sampling irrespective of longitude, only migrating components contribute to the diurnal cycle in our analyses". It is not clear why only migrating components contribute. " Response: The phases of nonmigrating tides at

different longitudes are different, so that the nonmigrating components average out if one averages over longitudes. We will mention this in the revised version.

3. "Figure 1a: In northern winter high OR near 15° latitude around 98 km is evident. What is the reason for that? What is the reason for equinoctial asymmetry in OR during SON while it is very symmetric during MAM?" Response: OR maximize in summer, but are also found in the winter hemisphere at low latitudes. Our Figure 1a is in accordance with climatologies, e.g. Arras et al. 2008. Actually the winter low-latitude maximum is also seen in the SH winter, but the values are lower because of the South Atlantic anomaly, where the Es OR are low, which leads so generally lower OR in the SH compared with the NH. Regarding the MAM and SON distribution, they are both not really symmetric about the equator, but the shapes of the respective summer and winter maximums are different. On the other hand, during both seasons the summer as well as the respective winter distributions look similar regarding their shape. The SON SH values are lower due to the South Atlantic anomaly. Interestingly, the MAM SH values are larger than the SON NH values which might be due to a hotspot over Indonesia and Australia, which is visible in the global distribution e.g. by Arras et al 2008. We will detail this in the revised version.

4. Figure 2: Only winter is shown. Results of all 4 seasons could be shown in order to be complete and consistent with Figure 1. Response: The figure is only shown to visualize the analysis, not to describe the diurnal wind shear. Actually the QDT component is hardly visibly in the left panel of figure 1, and this is also the case for the other seasons so that plotting these figures would not give more insight into the QDT behaviour.

5. Figure 4: Only a single representative altitude (110 km) is shown. More altitude bins (3 or more) could be included to strengthen the claim of the authors. Response: We selected 101 km because this is a MUAM level which is close to the maximum Es OR height. The lat/lon zonal wind amplitude distribution does not change much with altitude so that plotting the other height does not provide too much relevant information,

also having in mind that not the wind amplitude but the wind shear amplitude is the important parameter. However, we will add some more figures in a supplement in the revised version

6. Figure 6: There is a clear inconsistency in the QDT phase between the occurrence rate and wind shear in summer when QDT is found to be weak as mentioned by the authors. It implies that the estimated phases in the wind shear in summer are incorrect. Authors should substantiate the statement in detail. Response: We described this inadequately. Es and wind shear phases agree within their error bars even during summer. Actually, however, this is to a certain degree unexpected, because both Es and wind shear TDT are not significant. We will describe this more correctly and in more detail.

7. Figure 7: Only 110 km altitude is selected. At least 3 more altitude bins could be shown to corroborate the inference. Response: We show the vertical structure in Figure 8, so that we think more heights than 101 km in Figure 7 are not necessary. However, we will add more figures in a supplement.

---

## Author Response (AR1)

**Quarterdiurnal signature in sporadic E occurrence rates and comparison with neutral wind shear**

by Christoph Jacobi, Christina Arras, Christoph Geißler, and Friederike Lilienthal

Response to the reviewer´s comments

**Reviewer #1**

We are thankful for the referee's comments which helped us to considerably improve the paper. We addressed the raised issues in the revised version, and also corrected the technical points addressed. In response to the specific points, we will repeat each one here in italics and provide a reply below:

*1. P-4, Line. 12-13: Authors mentioned that "slightly above 100 km". Actually, it is mostly below 100 km.*

Response: We agree that our statement is only true for higher latitudes. We corrected that in the revised version.

*2. P-7, Line. 11-12: "Due to this sampling irrespective of longitude, only migrating components contribute to the diurnal cycle in our analyses". It is not clear why only migrating components contribute.*

Response: The phases of nonmigrating tides at different longitudes are different, so that the nonmigrating components average out if one averages over longitudes. We mention this in the revised version.

*3. Figure 1a: In northern winter high OR near 15◦latitude around 98 km is evident. What is the reason for that? What is the reason for equinoctial asymmetry in OR during SON while it is very symmetric during MAM?*

Response: OR maximize in summer, but are also found in the winter hemisphere at low latitudes. Our Figure 1a is in accordance with climatologies, e.g. in Arras et al. 2008. Actually the winter low-latitude maximum is also seen in the SH winter, but the values are lower because of the South Atlantic anomaly, where the Es OR are low. Regarding the MAM and SON distribution, they are both not really symmetric about the equator, but the shapes of the respective NH and SH maximums during one season are different. On the other hand, during both SON and MAM the autumn as well as the respective spring distributions look similar regarding their shape. The SON SH values are lower than the MAM NH due to the South Atlantic anomaly. Interestingly, the MAM SH values are larger than the SON NH values which might be due to a hotspot over Indonesia and Australia, which is visible in the global distribution e.g. by Arras et al 2008. We detailed this in the revised version.

*4. Figure 2: Only winter is shown. Results of all 4 seasons could be shown in order to be complete and consistent with Figure 1.*

Response: The figure is mainly shown to visualize the analysis, not to describe the diurnal wind shear. Actually, the QDT component is hardly visibly in the left panel of figure 1, and this is also the case for the other seasons so that plotting all of these figures would not give more insight into the QDT behaviour. We added and discussed, however, the summer wind and wind shear, because there the situation is different, when the positive mean wind shear in most cases dominate over the tidal shear.

*5. Figure 4: Only a single representative altitude (110 km) is shown. More altitude bins (3 or more) could be included to strengthen the claim of the authors.*

Response: We selected 101 km because this is a MUAM level which is close to the maximum Es OR height. The lat/lon zonal wind amplitude distribution does not change much with altitude so that plotting the other height does not provide too much relevant information, also having in mind that not the wind amplitude but the wind shear amplitude is the important parameter. However, we added some more figures in the appendix (Figs. A1-A3) to the revised version.

*6. Figure 6: There is a clear inconsistency in the QDT phase between the occurrence rate and wind shear in summer when QDT is found to be weak as mentioned by the authors. It implies that the estimated phases in the wind shear in summer are incorrect. Authors should substantiate the statement in detail.*

Response: We apologise for describing this inadequately. Es and wind shear phases agree within their error bars even during summer. Actually, however, this is to a certain degree unexpected, because both Es and wind shear TDT are not significant. We now describe this more correctly and in more detail.

*7. Figure 7: Only 110 km altitude is selected. At least 3 more altitude bins could be shown to corroborate the inference.*

Response: We show the vertical structure in Figure 8, so that we think more heights than 101 km in Figure 7 are not absolutely necessary. However, we added more figures in the appendix (Figs. A4-A6).

The technical points have been carefully addressed.

**Reviewer #2**

Thank you very much for your very positive opinion. Negative wind shear means that the vertical gradient of the considered horizontal wind component (either zonal or meridional, but we do not analyze the meridional one here) is negative, i.e. $dU/du < 0$ for negative zonal wind shear. We added this information in the revised version.

[revised manuscript text omitted]

---

## Author Response (AR2)

**Quarterdiurnal signature in sporadic E occurrence rates and comparison with neutral wind shear**

by Christoph Jacobi, Christina Arras, Christoph Geißler, and Friederike Lilienthal

Response to the reviewer #2 comments

Thank you very much for your positive view on our paper. We have corrected the indicated errors.

[revised manuscript text omitted]